# Influence of dietary composition on the nutritional profile and feed conversion efficiency of *Tenebrio molitor*

**Raziye Rashidi Ilzoleh, Vahid Akmali** *

Department of Biology, Faculty of Science, Razi University, Kermanshah, Iran

* v_akmali@razi.ac.ir

## Abstract

Insects, such as mealworm larvae, are promising sustainable protein sources due to their high reproductive ability, nutritional value, efficient organic matter conversion, cost-effective rearing, and minimal environmental impact. This study evaluates the nutritional composition and feed conversion efficiency of mealworm larvae reared on seven diets: W (100% wheat bran) as a control, A (50% wheat bran + 50% barley bran), B (75% wheat bran + 25% barley bran), C (50% wheat bran + 50% chickpea bran), D (75% wheat bran + 25% chickpea bran), E (50% wheat bran + 50% corn bran), and F (75% wheat bran + 25% corn bran). Results showed that diet C yielded the highest protein content and feed conversion efficiency, while diet B had the lowest. Fat content peaked in diets B and F. Variations in fiber, carbohydrates, ash, moisture, and minerals were also observed. Factors such as food intake, digestibility, and conversion ratios varied significantly among diets. The study highlights the critical role of dietary composition in optimizing mealworm larvae's nutritional profile and feed efficiency, offering a sustainable, cost-effective protein source for poultry and broilers. These findings support the strategic use of cereal brans to enhance feed quality, reduce costs, and improve scalability in insect farming, contributing to the sustainable production of animal protein.

## Introduction

The current sources of animal protein such as livestock, poultry, and fish, contribute significantly to environmental damage during production. On the other hand, the increasing trend in population growth leads to the demand for more food, particularly protein [1–5]. Insects with high reproductive ability, efficient organic matter conversion, and cost-effective rearing with high nutritional value and minimal greenhouse gas emissions, can be used as a viable alternative to traditional animal protein sources [2,6–8]. Insects exhibit rapid reproduction and play a crucial role in recycling organic waste while providing highly nutritious protein. Furthermore, insect farming

**Data availability statement:** all relevant data are within the manuscript and its Supporting information files.

**Funding:** This study received no external financial support from any organization. It was entirely funded by Razi University as part of an MSc thesis. Razi University's involvement was limited to the standard academic supervision of an MSc candidate and did not influence the study design, data collection and analysis, decision to publish, or preparation of the manuscript.

**Competing interests:** The authors have declared that no competing interests exist.

offers environmental advantages, including reduced land and water use, lower greenhouse gas emissions, and increased food efficiency [5,6,9]. Edible insects are often better than traditional protein sources due to their valuable nutritional content [3,10]. Globally, more than 2300 insect species are consumed by approximately 2.5 billion people, with the order Coleoptera being the most popular, comprising 31.2% of these species [11].

Among the numerous species suitable for mass production, *Tenebrio molitor* Linnaeus 1758 (Coleoptera: Tenebrionidae) is particularly notable due to its ease of rearing, low production cost, high nutritional value, rapid reproduction, and growth [5,12,13]. Its life cycle comprises four stages: egg, larva, pupa, and adult insect [13,14]. During the larval stage, the insects undergo multiple molts (between 9 and 23 times), changing from white to yellow, and increase in size [11,15,16]. *Tenebrio molitor* is not only nutritionally rich but also has the least environmental impact, requiring minimal land for rearing [5,11,16]. The larvae of *Tenebrio molitor*, commonly known as mealworms, are very popular due to their high nutritional value for feeding captive mammals, birds, reptiles, and amphibians [9,17]. These larvae serve as a stable and nutritious protein source, containing eighteen essential amino acids, as well as vitamins and minerals, making them suitable for both human consumption and animal feed [18]. Due to the rising costs of producing traditional protein sources such as soy and fishmeal, and the significant nutritional value of mealworms, they are increasingly being considered as an alternative protein source for broilers and poultry [12]. Additionally, these insects, known for their environmental compatibility and straightforward rearing, have the potential to recycle waste and provide food for a variety of animals [9,17,18].

The chemical composition and nutritional value of mealworms depend on the type of diet and environmental conditions in which they are reared. The nutritional composition of *Tenebrio molitor* larvae is influenced by their diet. Larvae fed high-protein diets exhibit elevated protein content [19]. A study investigating the use of plant waste as a substrate for *T. molitor* larvae demonstrated successful growth, with larvae achieving dry and fresh weight increases of 56.15% and 46.76%, respectively. The protein and fat contents of these larvae constituted 76.14% and 6.44% of the dry weight, respectively. Notably, larvae fed on plant waste exhibited lower fat content compared to those consuming wheat bran [20].

Another investigation explored the chemical composition of *T. molitor* larvae fed a diet containing palm (*Acrocomia aculeata*) powder, revealing that mealworms reared on this diet were a rich source of protein (44.83%), lipid (40.45%), and fatty acids (65.99%) [17]. In a study, researchers compared the composition of Korean and Chinese mealworms, considering general components such as moisture, crude protein, crude fat, crude ash, crude fiber, and carbohydrates. The analysis revealed that both Korean and Chinese mealworms were abundant in crude protein, with the protein content in Chinese mealworm powder slightly higher. The amino acid composition was similar, while the fatty acid profile differed between the two types of mealworms [21]. The type of feed and diet directly impact the composition of mealworm larvae, with existing research primarily focusing on protein and fat profiles, while mineral

profiles have received comparatively less attention. Recent studies indicate that specific minerals can be enhanced in mealworms by incorporating certain diets [22].

Insect farming offers numerous environmental benefits, primarily due to insects' high conversion efficiency and lower resource requirements compared to traditional livestock farming. The high ability to convert organic matter into biomass with high nutritional value in insects compared to livestock such as cattle causes a significant decrease in environmental impact [11,23]. Insects grow well on various organic materials, including organic waste, which can help reduce environmental pollutants by reusing these materials as valuable protein sources [11]. In contrast to traditional livestock, insects emit fewer greenhouse gases and less ammonia, contributing to a reduction in the effects of climate change [24].

Additionally, insect farming requires less land and water compared to raising larger animals such as cattle, making it a more efficient use of resources [11,23]. Given the distinct taxonomic differences between insects and humans as compared to traditional livestock, the risk of common infections between humans and insects is considerably lower [11]. The cold-blooded nature of insects, in contrast to livestock, contributes to their remarkable conversion efficiency. Insects, especially when compared to larger animals like cattle, exhibit a heightened ability to convert plant biomass into animal biomass [16,25]. Insects use the environment to regulate body temperature, consuming less energy to maintain body heat compared to warm-blooded animals, contributing to their higher food conversion efficiency [17,24–26]. The consumed diet plays a crucial role in determining factors such as feed conversion efficiency, growth rate, and the nutritional value of the insect body [24].

Globally, the demand for animal products has gradually increased. Insects, due to their high nutritional value, significant environmental benefits, and various economic considerations, are a suitable option for human food and animal feed [27]. Recent studies have explored the potential of mealworms as a sustainable protein source, investigating various substrates for rearing. While wheat bran remains a common substrate, alternatives like fresh plant materials, fungi and probiotics, and agricultural by-products have shown promise. These substrates can influence growth rates, nutritional profiles, and production efficiency. These findings highlight the potential for optimizing mealworm production using diverse, sustainable substrates [1,28,29]. Alternative bran sources from barley, corn, and chickpea may offer distinct nutritional profiles that could affect the larvae's growth performance, overall nutritional value, and nutrient absorption. Given the limited comprehensive studies on these alternative substrates, this research hypothesizes that incorporating barley, corn, and chickpea bran into the diet of *T. molitor* larvae will significantly impact their growth rates, nutritional composition, and nutrient absorption compared to a standard 100% wheat bran diet. Additionally, it is expected that optimal combinations of these alternative brans will enhance the larvae's performance and nutritional outcomes. This study aims to assess the impact of various substrates on the production, feed conversion efficiency metrics, and the nutritional and mineral composition of mealworm larvae.

## Materials and methods

### Mealworm rearing and diet preparation

The *T. molitor* larvae were obtained from the Ecology laboratory at the Department of Biology, Faculty of Science, Razi University, Kermanshah province, Iran. The larvae were then transferred to an insect breeding room at 28°C±1°C and a relative humidity of 65% (±5%) with a 0 L: 24 D photoregime. This controlled setup included 21 plastic containers measuring 15 cm x 28 cm x 22 cm [20,24,30–35]. To investigate the impact of various cereal bran on the nutritional composition and mineral content of mealworm larvae, seven different diets were provided for larval feeding including 100% wheat bran as a control (W), 50% barley bran +50% wheat bran (A), 25% barley bran +75% wheat bran (B), 50% chickpea bran +50% wheat bran (C), 25% chickpea bran +75% wheat bran (D), 50% corn bran +50% wheat bran (E), 25% corn bran +75% wheat bran (F). Before use, the ingredients (wheat bran and other grains) were mixed, homogenized, and stored at 4°C for 24 hours to reduce fungal and microbial contamination in the mealworm medium [20]. Newly hatched larvae were

allowed to feed ad libitum on each diet for four weeks before the experiment began. The experimental period was chosen because mortality among newly hatched larvae was higher than in later larval stages and growth rates can vary considerably between individual larvae. Three replicate containers were set up for each diet. After the initial period, fifty larvae were randomly transferred to each plastic container (22 × 28 × 15 cm) with aeration slits in the side [20,24,27,28,31,32]. For moisture, two grams of fresh carrot was added twice a week, and old carrot pieces were removed [24,26,34].

For chemical and nutritional analysis, 10-week-old larvae were harvested by sieving [36]. Prior to analysis, mealworms underwent a 24-hour starvation period to clear their gastrointestinal tracts, reducing microbial load and improving safety. After harvest, larvae were rinsed with tap water to remove debris, ensuring compliance with quality and safety standards for human consumption. Euthanasia was performed by freezing at −20°C, a method that minimizes stress by slowing metabolic activity and is ethically preferred over alternatives like boiling or crushing. This approach aligns with international research guidelines and industry standards for edible insect production [12,17,20,30–33].

## The nutritional composition

**Moisture content.** The moisture content was determined gravimetrically using 50 larvae (five from each weight group). The oven dry method procedure described in APHA [34] was followed. The larvae were first weighed (in groups of five larvae) using a a laboratory balance (RADWAG, WTC 200). They were then dried in a convection oven (Isotemp oven, Model No. 655F, Fisher Scientific, Montreal, Quebec) for 24 h at 105°C. The dried larvae were then removed from the oven, left to cool in a desiccator, and weighed. The moisture content was calculated according to Equation (1) [33]:

$$MC = \frac{M_1 - M_2}{M_1} \times 100$$

(1)

where $M_1$= weight of live larvae (mg), $M_2$= weight of dried larvae (mg), $MC$= percentage of moisture (%).

**Ash content and speciation of mineral elements.** The assessment of crude ash and mineral contents in the mealworm sample was conducted following the methods outlined by the Association of Official Analytical Chemists (AOAC, 1990). Crude ash content (CA) was determined by incineration samples in a muffle furnace at 550°C for four hours and measuring the weight loss. To prepare a solution containing mineral elements, two grams of the sample were placed into a crucible with ±1 mg accuracy and incinerated in an electric furnace at 550°C ± 20°C until white or gray ash formed. Five milliliters of concentrated nitric acid were then added to the ash, the crucible was covered with an aluminum lid, and the mixture was left at room temperature for one hour. The samples were heated in a Bain-Marie at 90°C for three hours and then cooled. Afterward, the contents were filtered using Whatman 42 filter paper with double-distilled water. The filtered solution was diluted to a final volume of 50 ml with deionized water, preparing it for analysis by atomic absorption spectroscopy. Phosphorus was measured by the molybdate vanadate method. Ammonium molybdate and vanadate reagent were added to the digested sample to form a yellow phosphomolybdate complex. Absorbance was then measured with a spectrophotometer at a wavelength of 400–430 nm, and phosphorus concentration was determined using a standard curve [19,21,22,32–35,37].

**Crude protein, Crude fat, carbohydrates, and fiber analyses.** Crude protein content was determined using the standard Kjeldahl method (AOAC Official Method 2001.11). This method involves measuring the nitrogen content of the sample and multiplying it by a factor of 6.25 to estimate the protein content. Crude fat (CFa) content was determined using petroleum ether extraction (AOAC Official Method 991.36). The crude fibre (CF) was quantified according to Commission Regulation No. 152/2009 [36,38]. This method involved the acid hydrolysis of one gram of sample with 150 mL of 0.13 mol/L H2SO4 at boiling point for five min for the extraction of sugars and starch, followed by alkaline hydrolysis with 150 mL of 0.23 mol/L KOH to remove proteins, hemi-cellulose, and lignin. The residue is filtered, dried, weighed, and ashed at 500°C for 1h. A blank test without sample was also conducted. The crude fibre content was calculated according to Equation (2):

$$\textbf{Crude Fibres} \left(\frac{\textbf{g}}{\textbf{100g}}\right) = (\textbf{m}_0 - \textbf{m}_1) \times \frac{\textbf{100}}{\textbf{m}}, \tag{2}$$

where m is the weight of the sample (g), $m_0$ is the weight lost after washing (g), and $m_1$ is the weight lost after ashing during the blank test (g).

The carbohydrate content was calculated by applying Equation (3) [36,39]:

$$\textbf{Carbohydrates} \left(\frac{\textbf{g}}{\textbf{100g}}\right) = \textbf{100} - (\textbf{protein} + \textbf{fat} + \textbf{ash} + \textbf{fiber}) \tag{3}$$

The energy provided by the samples (kcal/100 g) was calculated by considering Annex XIV of Regulation (EU) No 1169/2011 [33], as in Equation (4):

$$\textbf{Energy value} \left(\frac{\textbf{kcal}}{\textbf{100g}) = 4 \times \textbf{Carbohydrate}(}\%\right) + \textbf{4} \times \textbf{Protein}\,(\%) + \textbf{2} \times \textbf{Ffibre}\,(\%) + \textbf{9} \times \textbf{Fat}\,(\%). \tag{4}$$

### The feed conversion efficiency

**Experimental setup.** To determine diet consumption, a separate batch of larvae was allowed to fully consume the diet and carrot. Larvae were allowed to feed undisturbed for four weeks. Four-week-old larvae were selected for the experiment after sieving them to include fourth to sixth instar stages [34]. After this period, for each replicate, 50 larvae were randomly selected, weighed, and transferred to a plastic container (22 x 28 x 15 cm) with ventilation slits on the side [20,24,26,31,34,35]. Then, the larvae's weight was monitored as a group until 50% of the surviving larvae had pupated [26]. Throughout the experiment, 2 grams of carrots were provided twice a week. Any uneaten carrot was removed and dried at 100°C until reaching a constant weight. This dried weight was then compared to the dry weight of a carrot piece with the same initial fresh weight, cut from the same carrot used in the experiment [24,26,34].

**Monitoring and data collection.** Before the diet was completely consumed (determined based on visual observation of diet and feces), larvae were transferred to a container with fresh diet and carrot [26]. After harvesting the larvae, the residual mixture of leftover diet and feces was removed. The frass was separated using a 40-mesh screen and weighed, and the dry weight of the remaining feed material was then measured [20,26]. Diets, pure feces, and residues were dried at 100°C to a constant weight. Feed conversion efficiency was calculated on a dry matter basis as the Efficiency of Conversion of Ingested Food (ECI) [26]. At the end of the experiment, larvae were starved for 24 hours and then euthanized by freezing at −20°C. The larvae were dried using the procedures outlined in the earlier experiment [26]. The experiment lasted 12 weeks, divided into four 3-week periods. Food consumption and larval weight (at the beginning and end of each period) were tracked. At the end of the experiment, food consumption and food utilization parameters were calculated. The efficiency of food conversion was calculated based on live and dry weight, food consumed, and frass (feces) produced [26,34]. The initial and final weight of fresh and dry larvae, frass, and leftover food were measured using a laboratory scale with an accuracy of 0.001 grams. Food intake was calculated by subtracting the amount of unconsumed food from the total amount provided. The amount of food absorbed was determined by subtracting the weight of consumed food from the weight of frass [40]. Live weight gain (LWG) was determined by subtracting the initial larval weight from the cumulative weight of live larvae in each diet. The dry weight gain (DWG) was calculated by multiplying the live weight gain (LWG) by the reported dry weight ratio of *Tenebrio molitor* larvae (0.38) [34,41]. Food consumed (FC) was calculated by subtracting the weight of the remaining food from the weight of the food provided. Food assimilated (FA) was calculated by subtracting the frass weight from the weight of the FC [34,42]. Additionally, the measurement of larvae weight change was carried out in terms of the relative growth rate (RGR), during a feeding period according to Equation (5):

$$RGR = \frac{[Ln(W_f) - Ln(W_i)]}{T}, \tag{5}$$

where $W_f$ is the final weight and $W_i$ is the initial weight after time $T$ [43].

Dry weight ($DW(mg)$) is calculated from $FW$ and moisture content, given by using Equation (6):

$$DW = FW(1 - MC\%), \tag{6}$$

The economic coefficient ($EC$) of larval growth is calculated using Equation (7),

$$EC = (m_3/m_4) \times 100, \tag{7}$$

where $m_3$ is the increase of larval mass ($DW$), $m_4$ is the mass of feedstuff ($DW$) consumed at the same time [20].

Various parameters, including $FC$, $FA$, Approximately Digestibility ($AD$), conversion efficiency of ingested food ($ECI$), conversion efficiency of digested food ($ECD$), relative consumption rate ($RCR$), and food consumption rate ($FCR$), were calculated using Equations (8–13), respectively: [34,41,42].

$$FC = \text{The weight of the food provided} - \text{The weight of the remaining food}$$

$$FC = FC - \text{Weight frass} \tag{8}$$

$$AD = W_i - \text{Weight frass}/W_i \times 100 \tag{9}$$

$$ECI = \frac{DWG \times 100}{FC} \tag{10}$$

$$ECD = \frac{DWG \times 100}{FA} \tag{11}$$

$$RCR = W_i/DW \times T \tag{12}$$

where $W_i$ is the dry weight of food eaten by a larva, $DW$ is the dry weight of the larva, T is the feeding period.

$$FCR = W_i/DW \tag{13}$$

The results are presented as mean ± SD (n = 3). Data were analyzed using one-way ANOVA, followed by Tukey's post-hoc test, at a significance level of 0.05 for between-group comparisons.

The permit for working with live animals was issued with the approval of the Razi University Animal Ethics Committee, under the code number IR.RAZI.REC.1399.072.

## Results and discussion

In this study, we examined how different ratios of wheat bran to barley, corn, and chickpea bran affect the chemical composition and feed absorption efficiency of mealworm larvae. Diet plays an essential role in the growth, development, and overall performance of insects, significantly influencing parameters such as survival, reproduction, and metabolic

efficiency. The composition and quality of an insect's diet can lead to considerable variations in growth rates and feed conversion efficiencies [19,26,44,45]. A recent comprehensive review has summarized the various diets and protein sources evaluated for rearing *T. molitor,* providing valuable insights into the nutritional requirements and preferences of this species [13].

## Protein and fat content

The results of the study revealed that diet C yielded the highest protein content in the larvae, whereas diet A resulted in the lowest protein content (Fig S1 in S1 File, Table 1). Chickpea bran, with 23.5% protein content, emerged as the optimal protein source, whereas barley bran, with only 12% protein, resulted in lower larval protein percentages. These results underscore the importance of incorporating high-protein substrates into mealworm diets. Larvae fed diet B exhibited the highest fat content, while those diet E had the lowest. These results indicate that the choice of cereal bran significantly impacts the nutritional profile and growth efficiency of the larvae. Our findings align with those reported in previous studies [19,43,46], indicating that protein and fat levels in diets significantly influence the nutritional composition of larvae. The fat content in wheat bran, chickpea bran, and corn bran varies significantly [47], affecting both the fat composition of larvae and their growth efficiency. Other studies also highlight the importance of specific substrates, such as lentil flour and olive pomace, in enhancing protein and fat content in larvae [16,48]. Vegetable waste supplements, including cucumber and tomato byproducts, have been shown to enhance protein content and increase polyunsaturated fatty acids (PUFAs) [44]. The choice of food substrates for rearing *T. molitor* larvae significantly affects their growth, safety, and nutritional value [45]. The nutritional content of the larvae varied widely, with protein levels ranging from 44.1% to 51.8%, fat content between 28.6% and 34.8%, and fiber content from 10.5% to 14.9% [45]. However, the study also revealed that the heavy metal content in the larvae varies with their diet, indicating that not all substrates may be safe. This emphasizes the importance of monitoring and managing the substrates used in insect farming [45].

Crude protein levels in mealworms typically range from 15.5% to 19.7% [38]. Various agricultural by-products, such as dry cabbage, potato, wheat bran, and rice bran, can significantly influence these nutritional profiles [49]. It has been emphasized that dietary fats and carbohydrates affect larval composition more than protein content [49]. Additionally,

**Table 1. The nutritional composition of *Tenebrio molitor* larvae in seven diets.**

| Nutritional composition | Diets | | | | | | |
|---|---|---|---|---|---|---|---|
| | W | A | B | C | D | E | F |
| Protein% | 22.33 | 21.18 | 19.18 | 38.13 | 28.68 | 26.06 | 23.46 |
| Fat% | 14.00 | 16.00 | 19.00 | 17.00 | 16.00 | 16.00 | 12.00 |
| Fiber% | 4.60 | 5.10 | 5.00 | 4.70 | 5.40 | 6.30 | 6.10 |
| Carbohydrate% | 14.98 | 12.90 | 10.17 | 5.02 | 8.91 | 12.67 | 16.21 |
| Ash% | 1.17 | 1.34 | 1.31 | 1.40 | 1.42 | 1.34 | 1.25 |
| Moisture% | 52.12 | 53.68 | 55.34 | 43.15 | 50.39 | 50.23 | 53.18 |
| Zn (ppm) | 5.99 | 6.10 | 6.85 | 6.43 | 5.79 | 6.39 | 6.09 |
| Cu (ppm) | 1.17 | 1.08 | 1.20 | 1.24 | 1.29 | 1.19 | 1.13 |
| Ca (ppm) | 30.26 | 34.67 | 36.67 | 32.75 | 35.21 | 41.96 | 36.66 |
| Fe (ppm) | 2.58 | 3.02 | 3.11 | 2.61 | 2.48 | 2.79 | 2.90 |
| K (ppm) | 348.94 | 373.13 | 374.66 | 380.42 | 357.39 | 360.46 | 372.17 |
| P (ppm) | 183.12 | 161.10 | 191.10 | 164.64 | 182.28 | 183.12 | 205.8 |

Diet abbreviations: A (50% wheat bran+50% barley bran), B (75% wheat bran+25% barley bran), C (50% chickpea bran+50% wheat bran), D (25% chickpea bran+75% wheat bran), E (50% corn bran+50% wheat bran), F (25% corn bran+75% wheat bran), and W (100% wheat bran). Concentration (ppm).

potato waste diets have been found to increase fat content while decreasing protein levels compared to wheat bran [50]. Studies have reported high protein and fat content in mealworms, highlighting their potential as a sustainable protein source that can be further optimized through dietary modifications [51–53]. Overall, these findings underscore the potential for dietary optimization to enhance the nutritional value of *T. molitor* larvae while ensuring growth efficiency and safety.

### Fiber, Carbohydrates, Moisture, Ash and Energy values

**Fiber content.** The fiber levels were observed to be lowest in diet W and highest in diet E, fiber is essential for maintaining digestive health and nutrient absorption, and these variations reflect the impact of dietary components on digestion processes. This observation is consistent with studies that emphasize the variability of fiber content due to dietary differences. For instance, Baena and Cardona (2012) found that the type of substrate significantly influences fiber levels, with cocoa seed husks containing a high soluble fiber content of 8.66% [51]. Similarly, crude fiber levels of 6.1% were reported in mealworm larvae, reinforcing our findings on fiber content variability across different diets [53]. The higher fiber levels in corn bran-based diets suggest potential benefits for digestive health and nutrient absorption in mealworm larvae.

**Carbohydrate content.** Carbohydrate content varied considerably, with diet C containing only 5.02% carbohydrates in contrast to 16.21% in diet F (Fig S1 in S1 File, Table 1). It has been reported that mealworms naturally contain approximately 11.5% carbohydrates, highlighting the significant influence of diet on this composition [54]. Our findings complement this observation, demonstrating that dietary diversity, particularly with chickpea, enhances carbohydrate levels.

**Moisture and ash content.** Studies indicate that live mealworms contain approximately 62% moisture, significantly reducing their energy density compared to dried mealworms, which have around 5% moisture. This substantial difference emphasizes that live mealworms are considerably less energy-dense, while dried mealworms provide a concentrated source of nutrients resulting from moisture removal [51,55]. Our findings suggest that diets with a higher proportion of chickpea bran significantly influence larval moisture content. This observation aligns with research establishing a relationship between dietary amino acid concentration and larval moisture levels, noting that diets with lower amino acid concentrations often result in higher moisture content [56]. This correlation may help explain the moisture variation observed between chickpea-based and wheat bran diets. Additionally, the critical role of moisture content in determining the metabolic efficiency and overall quality of mealworms has been highlighted [44].

Moisture plays a fundamental role in enzymatic activities, nutrient absorption, and physiological processes, all of which impact larval development and quality. A reference range for the moisture content of mealworms has been established, indicating values between 60.2% and 74.8%, which helps contextualize the observations made in the Finke study [41]. Ash content remained relatively uniform, ranging between 1.17% and 1.42%, whereas moisture content was highest in diet B and lowest in diet C (Fig S1 in S1 File, Table 1). Additionally, much lower moisture content (4.75%) and ash content (4.125%) in mealworms have been reported, potentially due to differences in rearing conditions, diets, or processing methods [53]. These comparisons underscore the variability in the nutritional composition of mealworms influenced by diet formulation and environmental factors.

**Energy content.** The energy values for various diets labeled W, A, B, C, D, E, and F, measured in kilocalories per 100 grams, are as follows: W (284.04 kcal), A (290.52 kcal), B (280.40 kcal), C (335.00 kcal), D (305.16 kcal), E (313.68 kcal), and F (278.88 kcal). The average energy value among these diets is approximately 298.24 kcal per 100 grams. Diet C has the highest energy value at 335.00 kcal, indicating it is the most calorie-dense option, potentially beneficial for specific dietary needs. In contrast, diet F has the lowest energy value at 278.88 kcal, making it a better choice for individuals looking to reduce caloric intake. Furthermore, the range between the maximum and minimum values is 56.12 kcal, highlighting the variation in energy content across these diets (Fig S2 in S1 File). The energy content of live mealworms has been reported to average 206 kcal per 100 grams, although this value exhibits considerable variability depending on

the rearing diet [57]. This variability highlights the importance of carefully understanding and managing mealworm diets to optimize their energy content [57]. The highest energy value in diet C (335 kcal/100 g) suggests that chickpea-based diets may provide superior energy sources for larvae. These results demonstrate the significant impact of dietary composition on mealworm caloric content. The existing literature further supports these findings. For instance, it has been found that mealworms nourished with a diet consisting of 75% Irish potato waste achieved an elevated energy content of 598 kcal/100 g [50]. Furthermore, energy values of *T. molitor* meal were reported to exceed those of tilapia meal when offered to broiler chickens [58]. Similarly, energy values ranging from 700 to 800 kcal/100 g were documented in larvae reared on diets of wheat bran or sprouted potatoes [59]. These variations emphasize the significant impact of dietary substrates on energy density.

**Mineral composition.**  The larvae demonstrate excellent mineral content, positioning them as a valuable source of essential nutrients [28]. Similarly, research indicates that mealworms are notably rich in essential minerals, including magnesium, zinc, and iron [60]. In our study, the mineral analysis of mealworm larvae diets reveals significant variations in mineral content based on substrate composition. Notably, zinc (Zn) content was highest in Diet B and lowest in Diet D, while copper (Cu) reached its peak in Diet D but was lowest in Diet A. Calcium (Ca) was found to be highest in diet E and lowest in Diet W, indicating a clear advantage of certain substrates for enhancing calcium levels. Similar trends were observed for iron (Fe), with the highest levels in Diet B and the lowest in Diet D. Additionally, potassium (K) concentrations peaked in Diet C and were lowest in Diet W, while phosphorus (P) reached its highest levels in Diet F and its lowest in Diet A (Fig 1). These findings underscore the critical importance of precise dietary selection and balance to enhance the mineral content of mealworm larvae. In line with this analysis, it has been reported that the mineral content in larvae varies according to their diet, highlighting the necessity for careful monitoring to achieve optimal nutritional outcomes [45].

Further research supports that wheat-based diets lead to higher larval body weights in insects, while plant-based substrates, like turnip leaves and tubers, enrich macro and microelement content [5]. The mineral composition of edible insects, such as mealworms, varies significantly with dietary choices and processing methods [22]. Notably, poultry litter can serve as an effective, cost-efficient feed for *T. molitor*, with diets comprising 50% or more optimizing mineral uptake [61]. Dietary enrichment has been shown to significantly enhance mineral content; for instance, mealworms fed a high-calcium diet exhibited elevated calcium levels after a 48-hour period [62]. Comparatively, high-calcium cricket feed was found to maximize calcium content and calcium-to-phosphorus ratios in both mealworms and superworms (*Zophobas morio*) within the same timeframe [63]. Additionally, incorporating zinc in mealworm diets not only raises zinc levels but also reduces cadmium accumulation, thereby improving food safety. These findings indicate that targeted dietary modifications can efficiently improve the nutritional profile of insect larvae, potentially satisfying the calcium needs of insectivorous animals [64]. These findings highlight the necessity of selecting and balancing mealworm diets to optimize nutritional benefits while reducing the risk of toxic contaminants. This is especially important for ensuring the safety and effectiveness of mealworms as food or feed sources.

### Feed conversion efficiency analysis

The analysis of feed conversion efficiency includes three key metrics: ECI, ECD (Efficiency of Conversion of Digested food), and EC (Overall Conversion Efficiency). Diet C consistently exhibits the highest values, while Diet B shows the lowest. RCR, FA, and FC also vary significantly among diets, reaching their highest values in diet A and lowest in diet C. AD is highest in diet A and lowest in diet F. Additionally, the FCR is highest in diet B and lowest in diet C (Fig 2, Table 2). These findings indicate that different feeding factors or diets significantly influence the nutrient composition and transformation efficiency of mealworms.

The results showed that there was no significant difference between diet A and diet W in EC, FA, ECI, ECD, RCR and FCR parameters, but significant differences were observed between other diets. For the FC parameter, all diets showed a significant difference compared to the control diet. For the AD parameter, significant differences were observed in diets

A, F, and C compared to the control, while no differences were observed in the other diets. A notable difference in the RGR parameter was identified for diets C, D, E, and F compared to the control diet, whereas no significant distinctions were observed among the other diets (Fig 1, Table 1). The lowest and highest average values were observed for the following parameters: EC in diets B (12.87) and C (16.19); FA in diets C (21,152) and A (23,953); FC in diets C (17,209) and A (19,870.33); AD in diets F (80.69) and A (82.95); RGR in diets C (0.045) and A (0.047); ECI in diets B (12.87) and C (16.19); ECD in diets B (15.72) and C (19.91); RCR in diets C (1.69) and A (1.94); and FCR in diets C (6.17) and D (6.95). Nutrient conversion efficiency in insects is significantly influenced by dietary composition. Our study revealed notable variability in feed conversion efficiency metrics among the tested diets, with Diet C exhibiting superior performance in both ECI and ECD parameters. These findings are consistent with previous research, which emphasizes that factors such as protein and fiber content play a critical role in nutritional efficiency in insects [22].

Similar to the findings of previous studies, significant variations in the ECI and ECD were confirmed across different diets [26,65]. Among the diets tested, Diet B demonstrated the highest feed conversion efficiency (FCR), effectively converting feed into biomass. Conversely, Diet C exhibited the lowest FCR efficiency. These results underscore the crucial role of nutritional composition in optimizing ECI, with distinct feed substrates yielding varying conversion efficiencies. Our findings are consistent with prior research, emphasizing the critical role of diet in influencing mealworm growth

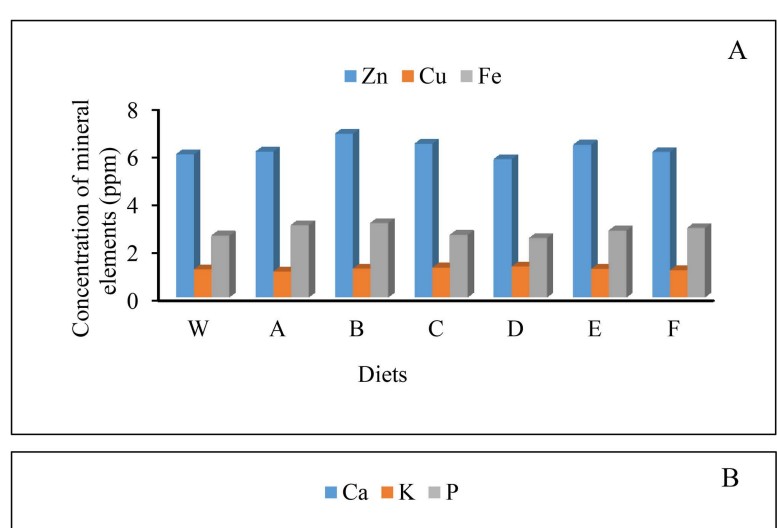

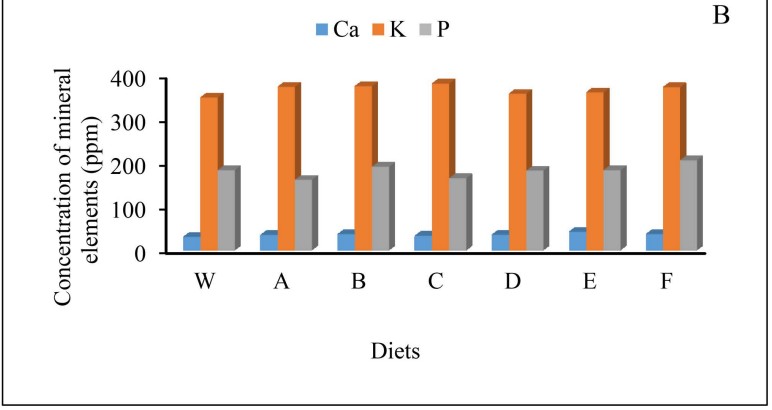

**Fig 1. Comparative chart showing the concentration of mineral elements including Zn, Cu, and Fe (A), Ca, K and P (B), in *Tenebrio molitor* larvae in seven experimental diets.** A (50% wheat bran + 50% barley bran), B (75% wheat bran + 25% barley bran), C (50% chickpea bran + 50% wheat bran), D (25% chickpea bran + 75% wheat bran), E (50% corn bran + 50% wheat bran), F (25% corn bran + 75% wheat bran), and W (100% wheat bran). Concentration (ppm).

performance and feed conversion efficiency, a key factor in bioconversion processes [22]. For example, oat flakes were found to achieve a lower feed conversion ratio (FCR), indicating greater efficiency compared to wheat bran [66]. This suggests that the enhanced performance of Diet B may reflect similar characteristics observed in that study. Additionally, significant differences in RGR were observed in diets C, D, E, and F compared to the control, while no distinctions were found among the remaining diets. The low RGR of Diet C is consistent with previous findings, demonstrating that dietary formulation significantly affects growth rates.

Absorbable Digestibility (AD) also varied across diets, with diets A, F, and C showing significant differences compared to the control. These variations emphasize the importance of nutrient utilization in growth performance. It was similarly reported that oat flakes exhibited superior digestibility compared to wheat bran, supporting the importance of digestibility in enhancing mealworm development [67]. Our study revealed significant differences in key parameters such as ECI, ECD, and RCR across the tested diets. These findings confirm that diet composition significantly influences the nutritional efficiency of mealworms. This highlights the potential for optimizing growth and feed conversion efficiency through carefully designed dietary formulations.

The research by van Broekhoven et al. 2015 emphasizes that variations in diet composition directly affect the digestibility and overall growth performance of larvae [26]. This aligns with the observed differences in absorbable digestibility (AD)

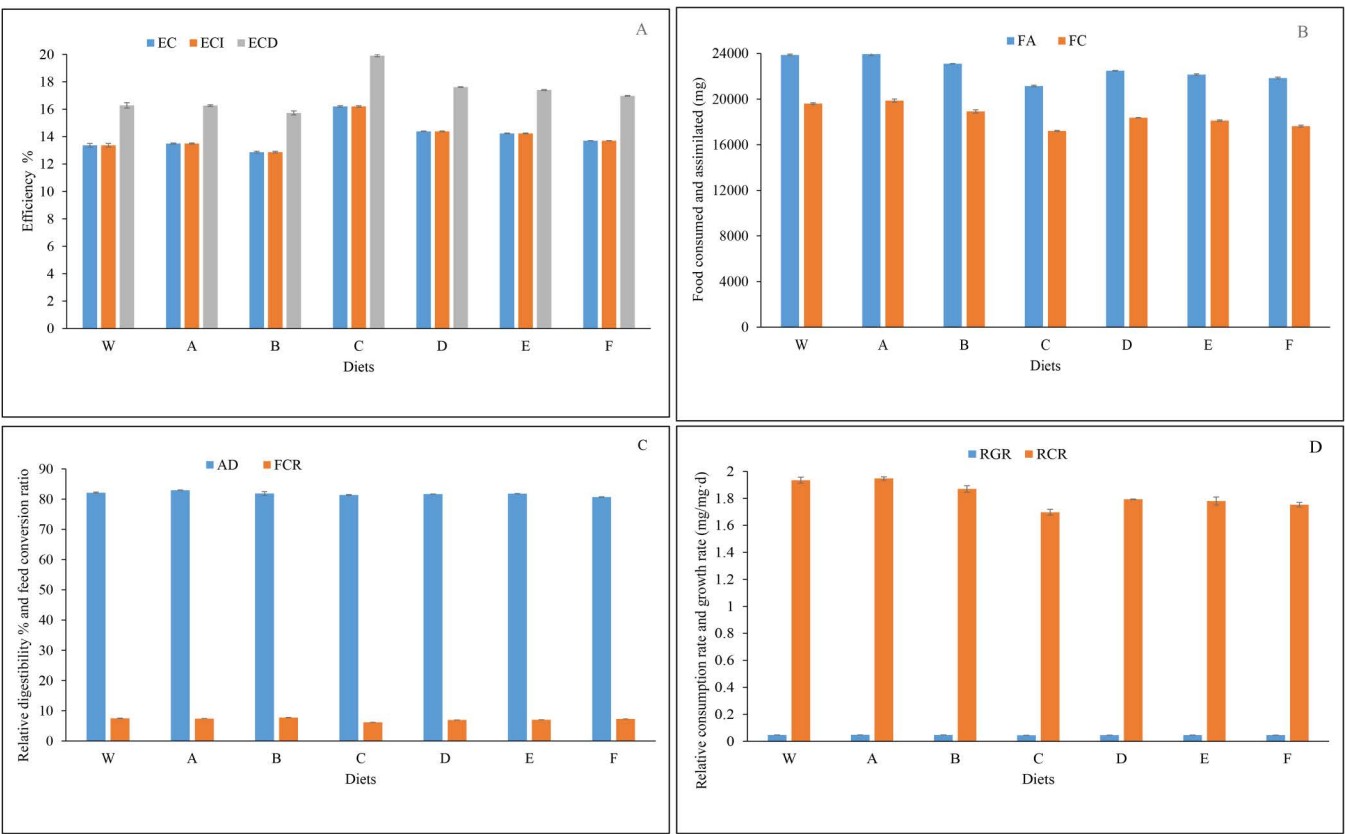

**Fig 2. Comparative chart of the food absorption parameters of the *Tenebrio molitor* larvae in seven experimental diets. A**: Percentage of conversion efficiency of ingested food (ECI), conversion efficiency of digested food (ECD), and economic coefficient (EC). **B**: Amount of food consumed (FC) and food assimilated (FA) (mg). **C**: Percentage of approximate digestibility (AD), and food consumption rate (FCR) (mg). **D**: Percentage of relative consumption rate (RCR) (% mg/d), and relative growth rate (RGR) (% mg/mg·d). Diet abbreviations: A (50% wheat bran + 50% barley bran), B (75% wheat bran + 25% barley bran), C (50% chickpea bran + 50% wheat bran), D (25% chickpea bran + 75% wheat bran), E (50% corn bran + 50% wheat bran), F (25% corn bran + 75% wheat bran), and W (100% wheat bran).

**Table 2. Food absorption (Mean±SD) parameters of *Tenebrio molitor* larvae in seven diets.**

| Food absorption | Diets | | | | | | | P-value | F-value |
|---|---|---|---|---|---|---|---|---|---|
| | W | A | B | C | D | E | F | | |
| EC (%) | 13.36±0.14 | 13.49±0.04 * | 12.87±0.06 | 16.2±0.06 | 14.39±0.02 | 14.24±0.03 | 13.7±0.01 | 0.00 | 857.307 |
| FA (mg) | 23877±68.42 | 23953±148.39 * | 23109±12.29 | 21152±69.09 | 22485.33±28.75 | 22144.67±67.99 | 21846±81.36 | 0.00 | 527.629 |
| FC (mg) | 19603±73.30 | 19870.33±138.82 | 18915.67±146.65 | 17209±39.15 | 18363.67±19.04 | 18115.33±64.26 | 17629.67±82.78 | 0.00 | 346.985 |
| AD (%) | 82.10±0.23 | 82.96±0.23 | 81.85±0.23 * | 81.36±0.23 * | 81.67±0.23 | 81.8±0.23 * | 80.7±0.23 * | 0.00001 | 21.753 |
| RGR(mg/mg·d) | 0.047±0.0005 | 0.048±0.00001 * | 0.047±0.0004 * | 0.045±0.0003 | 0.046±0.00001 | 0.046±0.0005 | 0.046±0.0002 | 0.00001 | 22.342 |
| ECI (%) | 13.36±0.138 | 13.49±0.039 * | 12.87±0.062 | 16.2±0.056 | 14.39±0.021 | 14.24±0.028 | 13.7±0.009 | 0.00 | 564.848 |
| ECD (%) | 16.28±0.1947 | 16.27±0.0571 * | 15.73±0.1449 | 19.91±0.0842 | 17.62±0.0203 | 17.4±0.0308 | 16.98±0.0245 | 0.00 | 564.848 |
| RCR(% mg/d) | 1.94±0.0218 | 1.95±0.0121 * | 1.87±0.0237 | 1.7±0.0209 | 1.79±0.0023 | 1.78±0.0294 | 1.75±0.0164 | 0.00 | 67.784 |
| FCR (mg) | 7.48±0.077 | 7.41±0.021 * | 7.77±0.037 | 6.17±0.021 | 6.95±0.010 | 7.02±0.014 | 7.3±0.005 | 0.00 | 654.167 |

*In the diets marked with an asterisk, no significant difference was observed with the control group.

Diet abbreviations: A (50% wheat bran+50% barley bran), B (75% wheat bran+25% barley bran), C (50% chickpea bran+50% wheat bran), D (25% chickpea bran+75% wheat bran), E (50% corn bran+50% wheat bran), F (25% corn bran+75% wheat bran), and W (100% wheat bran).

Legend: Food consumed (FC), food assimilated (FA), approximate digestibility (AD), economic coefficient (EC), conversion efficiency of ingested food (ECI), conversion efficiency of digested food (ECD), relative growth rate (RGR), relative consumption rate (RCR), and food consumption rate (FCR).

across the diets in our study. The results indicate that specific dietary formulations can lead to improved nutrient absorption, a critical factor for optimizing larval growth and enhancing feed efficiency. Our findings, particularly the improved conversion efficiency with chickpea bran, highlight how diet influences both growth metrics and biological processes in mealworms. The potential of mealworms as a sustainable protein source, particularly their ability to convert organic by-products into biomass, is supported by recent studies [31,50]. Our work further confirms that specific components, such as chickpea and barley bran, optimize growth and conversion rates, advancing sustainable insect production.

This study enhances our understanding of how specific dietary components influence feed conversion efficiency (FCE) in mealworms. It underscores the importance of strategically formulating diets to optimize the Efficiency of Conversion of Ingested Feed (ECI), Efficiency of Conversion of Digested Feed (ECD), and nutrient absorption, while also reducing the Feed Conversion Ratio (FCR). Tailoring dietary interventions can significantly enhance larval growth and efficiency, promoting more sustainable insect production and improved resource utilization in agriculture. Studies have shown that alternative feed sources, such as poultry litter, can support development and reproductive efficiency comparable to traditional diets, making them a viable and cost-effective option for sustainable rearing [61].

Studies have shown that replacing wheat bran with corn bran or poultry litter does not significantly affect larval development or weight, suggesting that alternative substrates can effectively replace conventional ones without compromising growth [61,67]. Additionally, it has been demonstrated that diet composition, particularly starch and protein content, plays a crucial role in growth and feed conversion, with higher starch levels improving conversion rates and protein-enriched diets enhancing growth [68]. Furthermore, incorporating Irish potato waste (PW) into *T. molitor* diets has been shown to promote greater weight gain and feed conversion efficiency, reinforcing the potential of using alternative, cost-effective substrates to improve mealworm production. These findings emphasize the promise of sustainable feed sources in supporting growth and nutritional quality in mealworm farming [50].

The significance of this study lies in its novel approach to evaluating the impact of alternative bran sources on mealworm nutrition and growth, thereby addressing an important gap in existing literature. While prior research has predominantly centered on the conventional use of wheat bran, this study explores the potential advantages and limitations of incorporating barley, corn, and chickpea bran. By systematically comparing these dietary components, it advances our understanding of how diverse substrates influence insect physiology and nutrient uptake. These findings are especially pertinent in light of the increasing interest in entomophagy and sustainable protein production. Despite its contributions, this research has several limitations. First, the study is limited by the specific types and ratios of bran used, which may not fully represent the potential effects of other grain by-products. Second, genetic variability among the larvae were not rigorously controlled, potentially affecting the generalizability of the findings. Nevertheless, the implications of this work are significant. Optimizing mealworm diets could enhance the efficiency of insect-rearing practices, improve the nutritional quality of the larvae, and support the development of more sustainable and economically viable protein sources. Future studies should explore a broader range of dietary formulations and investigate the underlying mechanisms driving the observed nutritional and physiological changes.

## Conclusions

The study demonstrates the potential of mealworm larvae as a sustainable protein source and highlights the importance of dietary composition in determining the nutritional profile and feed conversion efficiency. The use of various types of cereal bran significantly impacts both the food composition and feed conversion efficiency of *Tenebrio molitor* larvae. The findings of the study showed that diets containing 50% chickpea bran had the highest protein levels. Additionally, combinations of barley and corn bran were associated with increased fat content. Moreover, the highest conversion efficiency was observed in diets with 50% chickpea bran and 50% barley bran, indicating that these combinations optimize the conversion of food into biomass. The variations observed in the larvae's nutritional composition and feed conversion efficiency among different cereal bran types could imply differences in their nutrient profiles, fiber content, or other components that

affect the larvae's development and metabolic processes. This suggests that the choice of cereal bran as a substrate or food source plays a pivotal role in influencing the nutritional aspects and how efficiently the larvae convert this food into growth and energy. The findings emphasize the importance of carefully selecting and balancing the proportions of different cereal brans in the larval diet to optimize the larvae's nutritional quality and growth performance. This research provides valuable insights into optimizing larval diets to improve protein yield and overall health, contributing to the advancement of insect-based food sources as a sustainable alternative to traditional livestock. These insights can contribute to the development of more efficient and sustainable insect-based protein production systems.

## Supporting information

**S1 File. Fig S1 and Fig S2.**
(DOCX)

**S2 File. Complete raw data.**
(SAV)

## Acknowledgments

We thank the Razi University authorities, Kermanshah, for their help and support of this study as a part of the MSc thesis of the first author. We are especially grateful to the editor and anonymous reviewers who remarkably improved early version of the manuscript.

## Author contributions

**Conceptualization:** Raziye Rashidi Ilzoleh, Vahid Akmali.

**Data curation:** Raziye Rashidi Ilzoleh, Vahid Akmali.

**Formal analysis:** Raziye Rashidi Ilzoleh, Vahid Akmali.

**Funding acquisition:** Vahid Akmali.

**Investigation:** Raziye Rashidi Ilzoleh, Vahid Akmali.

**Methodology:** Raziye Rashidi Ilzoleh, Vahid Akmali.

**Project administration:** Vahid Akmali.

**Resources:** Vahid Akmali.

**Software:** Raziye Rashidi Ilzoleh, Vahid Akmali.

**Supervision:** Vahid Akmali.

**Validation:** Vahid Akmali.

**Visualization:** Vahid Akmali.

**Writing – original draft:** Raziye Rashidi Ilzoleh, Vahid Akmali.

**Writing – review & editing:** Vahid Akmali.

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
