## [Decision Letter · Decision Letter 0]

Dear Dr. Akmali,

Thank you for submitting your manuscript to PLOS ONE. After careful consideration, we feel that it has merit but does not fully meet PLOS ONE’s publication criteria as it currently stands. Therefore, we invite you to submit a revised version of the manuscript that addresses the points raised during the review process.

We look forward to receiving your revised manuscript.

Kind regards,

António Raposo

Academic Editor

PLOS ONE

**Journal Requirements:**

1. When submitting your revision, we need you to address these additional requirements. Please ensure that your manuscript meets PLOS ONE's style requirements, including those for file naming. The PLOS ONE style templates can be found at https://journals.plos.org/plosone/s/file?id=wjVg/PLOSOne_formatting_sample_main_body.pdf and https://journals.plos.org/plosone/s/file?id=ba62/PLOSOne_formatting_sample_title_authors_affiliations.pdf 2. In your Methods section, please provide additional information regarding the permits you obtained for the work. Please ensure you have included the full name of the authority that approved the field site access and, if no permits were required, a brief statement explaining why. 3. We note that your Data Availability Statement is currently as follows: All relevant data are within the manuscript and its Supporting Information files. Please confirm at this time whether or not your submission contains all raw data required to replicate the results of your study. Authors must share the “minimal data set” for their submission. PLOS defines the minimal data set to consist of the data required to replicate all study findings reported in the article, as well as related metadata and methods (https://journals.plos.org/plosone/s/data-availability#loc-minimal-data-set-definition). For example, authors should submit the following data: - The values behind the means, standard deviations and other measures reported;- The values used to build graphs;- The points extracted from images for analysis. Authors do not need to submit their entire data set if only a portion of the data was used in the reported study. If your submission does not contain these data, please either upload them as Supporting Information files or deposit them to a stable, public repository and provide us with the relevant URLs, DOIs, or accession numbers. For a list of recommended repositories, please see https://journals.plos.org/plosone/s/recommended-repositories. If there are ethical or legal restrictions on sharing a de-identified data set, please explain them in detail (e.g., data contain potentially sensitive information, data are owned by a third-party organization, etc.) and who has imposed them (e.g., an ethics committee). Please also provide contact information for a data access committee, ethics committee, or other institutional body to which data requests may be sent. If data are owned by a third party, please indicate how others may request data access. 4. Please include your full ethics statement in the ‘Methods’ section of your manuscript file. In your statement, please include the full name of the IRB or ethics committee who approved or waived your study, as well as whether or not you obtained informed written or verbal consent. If consent was waived for your study, please include this information in your statement as well.

Reviewers' comments:

Reviewer's Responses to Questions

**Comments to the Author**

1. Is the manuscript technically sound, and do the data support the conclusions?

Reviewer #1: Partly

Reviewer #2: Yes

Reviewer #3: Yes

Reviewer #4: No

Reviewer #5: Yes

2. Has the statistical analysis been performed appropriately and rigorously?

Reviewer #1: Yes

Reviewer #2: Yes

Reviewer #3: Yes

Reviewer #4: No

Reviewer #5: Yes

3. Have the authors made all data underlying the findings in their manuscript fully available?

Reviewer #1: Yes

Reviewer #2: Yes

Reviewer #3: Yes

Reviewer #4: Yes

Reviewer #5: Yes

4. Is the manuscript presented in an intelligible fashion and written in standard English?

Reviewer #1: No

Reviewer #2: No

Reviewer #3: Yes

Reviewer #4: Yes

Reviewer #5: Yes

**Reviewer #1: ** Respectfully

The study was statistically and content-wise reviewed.

Please make the following corrections:

It is unclear how many mealworms were used per treatment and whether replicates were biological or technical.

The manuscript states that three replicates per diet were used, but this may not be sufficient for robust statistical comparisons.

The study does not report mycotoxin screening or contamination control in dietary ingredients.

Given that bran-based diets can be prone to fungal or bacterial contamination, this omission is a potential flaw affecting validity.

While protein content is analyzed, no detailed amino acid profile is provided, which is crucial for evaluating the nutritional quality of mealworms.

The manuscript states that the study was approved by the Razi University Ethics Committee, but the specific ethical considerations for insect research are not discussed.

More details on ethical handling and humane euthanasia methods for mealworms should be included.

The conclusion claims broad applicability of the findings to mealworm farming, but these claims lack sufficient evidence from large-scale trials.

More discussion on real-world feasibility and potential economic implications of using chickpea bran in commercial mealworm farming is needed.

The discussion cites relevant literature, but comparisons with previous studies are weak.

It is unclear how the results align or contradict existing research on insect nutrition.

A more systematic comparison with past studies is needed to establish the novelty of the work.

The study evaluates mineral content, but it does not consider possible heavy metal accumulation in larvae, which is a critical concern for human and animal consumption.

A brief discussion of food safety risks would improve the manuscript.

The manuscript contains several grammatical errors and awkward phrasing that reduce readability.

Example:

"The results showed that diets with 50% chickpea bran and 50% barley bran had the highest and lowest protein content in larvae."

Revision: "Diets containing 50% chickpea bran produced larvae with the highest protein content, while those with 50% barley bran yielded the lowest."

Some figures (e.g., Fig. 1 and Fig. 3) lack clear labeling and legends.

Tables do not adequately summarize key statistical findings (e.g., missing p-values).

5. Recommendations for Improvement

1. Clarify the research question and hypothesis to provide a stronger framework for the study.

2. Improve statistical analyses, including effect size reporting, confidence intervals, and power analysis.

3. Justify diet selection based on scientific rationale rather than arbitrary choices.

4. Ensure that all feed conversion calculations account for moisture content to prevent misinterpretation of FCE results.

5. Provide amino acid profiling to enhance the discussion on nutritional quality.

6. Discuss potential biases and limitations more explicitly.

7. Improve clarity and organization of tables and figures for better readability.

8. Refine the discussion section to integrate findings with prior research more effectively.

9. Check for potential contamination in dietary substrates and discuss implications.

10. Address ethical considerations in insect research, including humane handling and euthanasia.

Use and cite the following studies to improve the structure and content of your study:

-A review of cultural aspects and barriers to the consumption of edible insects

-First Report of Hermetia Illucens (Linnaeus, 1758), Black Soldier Fly (Diptera, Stratiomyidae) from Iran

-The Effects of Curcumin Supplementation on Body Weight, Body Mass Index, and Waist Circumference in Patients with Type 2 Diabetes: A Systematic Review and Meta-Analysis of Randomized Controlled Trials

-Halal Certification for edible insects

The study needs a complete revision in terms of language and grammar.

Good luck.

**Reviewer #2: ** I have found the manuscript important and worthy of consideration. This study provides valuable insights on “Influence of dietary composition on the nutritional profile and feed conversion efficiency of Tenebrio molitor”. However, it requires minor revisions before the editor makes a final decision.

1. The abstract is quite long. You can consider making it more concise. Please clarify the practical applications of the findings, like implications for large-scale production.

2. Could you discuss in the introduction why this study is necessary compared to existing research?

3. In the method section, could you briefly explain why ANOVA and Tukey’s test were chosen?

One suggestion for language and formatting. While reading, I have found some sentences are overly complex, simplify those for readability.

Thank you.

**Reviewer #3: ** Introduction

• The research gap is not strongly justified. The introduction should explicitly state what is missing in current knowledge and why this study is needed.

• The introduction should mention any previous studies that have examined similar dietary influences on mealworms and explain how this study builds upon or differs from them.

• It lacks a theoretical framework for insect nutrition—key biological principles governing mealworm growth (e.g., metabolic efficiency, nutrient assimilation) could strengthen the justification.

• The hypothesis or specific research questions are not explicitly stated. A clear hypothesis would improve focus.

Methodology

• The study lacks a clear justification for the sample size. How was the number of mealworms per treatment determined? Was there a power analysis?

• The experiment lasted 12 weeks, but there is no discussion on whether this timeframe is sufficient to assess long-term dietary effects.

• Potential biases: Since mealworm growth can be influenced by factors such as rearing density, genetic variability, and microclimate conditions, were these controlled?

• The moisture addition (carrots) could introduce variation. Was this accounted for in data interpretation?

Results

• Some results lack clear interpretation—for example, what biological mechanisms might explain why diet C had the highest conversion efficiency?

• The statistical significance of certain comparisons is not always explained in a biologically meaningful way.

• Figures lack sufficient labeling—ensure all legends and axis labels are self-explanatory.

Discussion

• The discussion sometimes repeats results rather than interpreting them in-depth.

• The implications for real-world applications (e.g., poultry feed formulation, industrial insect farming) could be expanded.

• It lacks a "Limitations and Future Research" section, which is important for acknowledging study constraints.

**Reviewer #4: ** Dear Authors,

After reviewing your ms, I conclude that it is not ready to publish. I have three important parts, either are missing and must add, either must be revised.

1) The statistics are missing. You must add a statistic model to your results.

2) The results must be separated from the discussion because it is better to read you text.

3) The figures must be more readable.

4) You don’t answer at the discussion why this research is important

**Reviewer #5:**  The article "Influence of dietary composition on the nutritional profile and feed conversion efficiency of Tenebrio molitor" is an original work that brings relevant and detailed aspects about the diet of a species of larvae used in poultry and other animal farming. The methodology of the work is clear and well-conducted. Additionally, the presentation, writing, and discussion of the data are good. I have a few considerations for the authors that could enhance the quality of the work:

In some sections of the Results and Discussion, the authors mention data that are simultaneously shown in figures and tables, such as Figure 1 and Table 1. I believe this redundancy of information is unnecessary. I think Figure 1 could be moved to supplementary material. I suggest revising the tables and figures and keeping only one method of presenting results in the main text.

I suggest the authors use only the designations for the diets: A, B, C, etc. At several points, the authors return to mention the composition of the diets already described in the Methods.

The authors also cite details contained in the tables, such as averages and percentages, within the text, which makes it tiring to read. I suggest avoiding such repetitions.

There are inconsistencies related to the percentages of the diet, for example, between lines 300-301. I suggest reviewing the entire section.

**Do you want your identity to be public for this peer review?** For information about this choice, including consent withdrawal, please see our Privacy Policy

Reviewer #1: **Yes: ** EBRAHIM ABBASI

Reviewer #2: No

Reviewer #3: No

Reviewer #4: No

Reviewer #5: No

---

## [Author Response · Author response to Decision Letter 1]

6 Apr 2025

Response to Reviewer Comments

We sincerely appreciate the reviewers' insightful and constructive comments, which have greatly contributed to improving the quality and clarity of our manuscript. Addressing these comments has allowed us to refine our study and better highlight its significance. Below, we provide detailed responses to each of the reviewers' comments, outlining the specific revisions made to the manuscript. We are grateful for the time and effort the reviewers have dedicated to evaluating our work and hope that the revised version meets their expectations. Thank you for your valuable feedback and consideration."

Reviewer #1:

Comment 1: It is unclear how many mealworms were used per treatment and whether replicates were biological or technical.

The manuscript states that three replicates per diet were used, but this may not be sufficient for robust statistical comparisons.

Response Thank you for raising this important point. To clarify, each replicate consisted of fifty larvae, which were randomly assigned to a plastic container (22 × 28 × 15 cm) equipped with aeration slits on the sides. The three replicates per diet were biological replicates, meaning they represented independent groups of larvae raised under the same dietary conditions. We appreciate your feedback and have revised the manuscript to include these clarifications."

Comment 2: The study does not report mycotoxin screening or contamination control in dietary ingredients.

Given that bran-based diets can be prone to fungal or bacterial contamination, this omission is a potential flaw affecting validity.

Response Prior to use, the ingredients (wheat bran and other grains) were mixed, homogenized, and stored at 4°C for 24 hours to reduce fungal and microbial contamination in the mealworm medium.

Refrences: Li L, Zhao Z, Liu H. Feasibility of feeding yellow mealworm (Tenebrio molitor L.) in bioregenerative life support systems as a source of animal protein for humans. Acta Astronautica. 2013;92(1):103-9.

Comment 3: While protein content is analyzed, no detailed amino acid profile is provided, which is crucial for evaluating the nutritional quality of mealworms.

Response We appreciate this valuable observation. As this study is part of an MSc thesis, the scope was focused on analyzing the protein content of mealworms. However, we fully recognize the importance of evaluating the amino acid profile to comprehensively assess nutritional quality. Future research will expand on these findings by investigating the detailed amino acid composition of mealworms, as well as exploring potential heavy metal accumulation.

Comment 4: The manuscript states that the study was approved by the Razi University Ethics Committee, but the specific ethical considerations for insect research are not discussed. More details on ethical handling and humane euthanasia methods for mealworms should be included.

Response Thank you for comment. We have now included additional details regarding the ethical handling and humane euthanasia methods for mealworms in the revised manuscript. The mealworms underwent a 24-hour starvation period, during which their feed was removed. This practice is commonly employed to empty their gastrointestinal tracts, reducing microbial load and enhancing safety for human consumption. After harvest, the larvae were thoroughly rinsed with tap water to remove any residual debris. For humane euthanasia, the larvae were frozen at -20°C. Freezing at this temperature is widely recognized as an ethical method for euthanizing insects, as it minimizes pain and stress by gradually slowing their metabolic rate and physical activity. This approach is considered superior to alternative methods, such as boiling or crushing, and aligns with international research guidelines and standard practices in the edible insect industry (van Broekhoven et al., 2015).

Reference:

van Broekhoven S, Oonincx DGAB, van Huis A, van Loon JJA. Growth performance and feed conversion efficiency of three edible mealworm species (Coleoptera: Tenebrionidae) on diets composed of organic by-products. Journal of Insect Physiology. 2015;73:1-10.

Comment 6: The conclusion claims broad applicability of the findings to mealworm farming, but these claims lack sufficient evidence from large-scale trials. More discussion on real-world feasibility and potential economic implications of using chickpea bran in commercial mealworm farming is needed.

Response Mealworms (Tenebrio molitor larvae) are increasingly recognized as a sustainable protein source for both animal feed and human nutrition. Traditionally, wheat bran has been the primary dry food used to rear these insects. However, alternative bran sources, such as barley, corn, and chickpea, may offer unique nutritional profiles that could impact the larvae's growth performance, overall nutritional value, and nutrient absorption. Given the limited research on these alternative substrates, this study hypothesizes that incorporating barley, corn, and chickpea bran into the diet of T. molitor larvae will result in significant differences in their growth rates, nutritional composition, and nutrient absorption efficiency compared to a standard 100% wheat bran diet. Additionally, it is expected that specific combinations or proportions of these alternative brans may optimize the larvae's performance and nutritional outcomes.

Comment 7: More discussion on real-world feasibility and potential economic implications of using chickpea bran in commercial mealworm farming is needed.

Response Thank you for this valuable suggestion. While our findings indicate that chickpea bran shows promise as a dietary substrate for mealworms, it is important to emphasize that these results were obtained under controlled experimental conditions. To fully assess the real-world feasibility and economic viability of incorporating chickpea bran into commercial mealworm farming, additional large-scale trials are necessary.

Comment 8: The discussion cites relevant literature, but comparisons with previous studies are weak.

It is unclear how the results align or contradict existing research on insect nutrition. A more systematic comparison with past studies is needed to establish the novelty of the work.

Response Thank you for this insightful comment. We acknowledge that the discussion could benefit from a more systematic comparison with existing literature to better contextualize our findings. In the revised manuscript, we have strengthened the discussion by explicitly comparing our results with those of previous studies on insect nutrition.

Comment 9: The study evaluates mineral content, but it does not consider possible heavy metal accumulation in larvae, which is a critical concern for human and animal consumption.

A brief discussion of food safety risks would improve the manuscript.

Response We appreciate this important observation. While the current study focused on evaluating the mineral content of mealworms, we recognize that assessing heavy metal accumulation is crucial for ensuring the safety of mealworms as a food source for humans and animals. We emphasize that future research should incorporate a thorough analysis of heavy metal levels in mealworms reared on various diets, including chickpea bran, to address this critical gap.

Comment 10: The manuscript contains several grammatical errors and awkward phrasing that reduce readability.

Example:

"The results showed that diets with 50% chickpea bran and 50% barley bran had the highest and lowest protein content in larvae."

Revision: "Diets containing 50% chickpea bran produced larvae with the highest protein content, while those with 50% barley bran yielded the lowest."

Response Thank you for your feedback. We have carefully reviewed the manuscript and made extensive revisions to correct grammatical errors and improve awkward phrasing. For instance, the sentence you highlighted has been revised as follows: 'Diets containing 50% chickpea bran produced larvae with the highest protein content, while those with 50% barley bran yielded the lowest.' These changes, along with similar revisions throughout the manuscript, have significantly enhanced clarity, readability, and overall flow. We appreciate your attention to detail and hope the revised version meets your expectations."

Comment 11: Some figures (e.g., Fig. 1 and Fig. 3) lack clear labeling and legends.

Tables do not adequately summarize key statistical findings (e.g., missing p-values).

Response Thank you for your valuable feedback. According to one of the reviewers, Figures 1 and 3 have been moved to the supplementary section. In the revised manuscript, we have updated the figures to include clearer labels, legends, and annotations to ensure they are more intuitive and informative. Additionally, we have revised the table 2 to include all relevant statistical findings, such as p-values, to provide a comprehensive summary of the data.

Comment 12: Use and cite the following studies to improve the structure and content of your study:

-A review of cultural aspects and barriers to the consumption of edible insects

-First Report of Hermetia Illucens (Linnaeus, 1758), Black Soldier Fly (Diptera, Stratiomyidae) from Iran

-The Effects of Curcumin Supplementation on Body Weight, Body Mass Index, and Waist Circumference in Patients with Type 2 Diabetes: A Systematic Review and Meta-Analysis of Randomized Controlled Trials

Response Thank you for your suggestions. we have cited the study on the cultural aspects and barriers to the consumption of edible insects.

Reviewer #2:

I have found the manuscript important and worthy of consideration. This study provides valuable insights on “Influence of dietary composition on the nutritional profile and feed conversion efficiency of Tenebrio molitor”. However, it requires minor revisions before the editor makes a final decision.

Comment 1. The abstract is quite long. You can consider making it more concise. Please clarify the practical applications of the findings, like implications for large-scale production.

Response Thank you for your feedback. We have revised the abstract to make it more concise while retaining the key points of the study. Additionally, we have clarified the practical applications of the findings, particularly their implications for large-scale mealworm production.

Comment 2. Could you discuss in the introduction why this study is necessary compared to existing research?

Response Thank you for your valuable suggestion. In the revised introduction, we have emphasized the necessity of this study by highlighting gaps in existing research. Specifically, while previous studies have explored various feed formulations for mealworms, there is limited research on the use of cereal bran, such as chickpea bran, as a sustainable and cost-effective dietary substrate.

Comment 3. In the method section, could you briefly explain why ANOVA and Tukey’s test were chosen?

Response Thank you for your comment. ANOVA was chosen for its ability to compare means across multiple groups, allowing us to assess the overall effects of different bran diets on mealworm growth and nutrition. Tukey’s test was selected post-ANOVA for its effectiveness in identifying specific group differences while controlling for Type I error, ensuring robust and reliable comparisons among the bran treatments.

Comment 4. One suggestion for language and formatting. While reading, I have found some sentences are overly complex, simplify those for readability.

Response Thank you for your suggestion. We have revised the manuscript to simplify overly complex sentences, improving readability without compromising scientific accuracy.

Reviewer #3:

Introduction

Comment 1: The research gap is not strongly justified. The introduction should explicitly state what is missing in current knowledge and why this study is needed.

Response Thank you for your feedback. In the introduction, I have now emphasized the necessity of this study by highlighting gaps in existing research, particularly the limited exploration of cereal bran's role in optimizing feed formulations for mealworm production.

Comment 2 The introduction should mention any previous studies that have examined similar dietary influences on mealworms and explain how this study builds upon or differs from them.

Response Previous studies have primarily focused on wheat bran and other single-source diets for mealworms. However, limited research has compared the effects of mixed or alternative bran sources such as barley, corn, and chickpea on mealworm nutrition and growth. By incorporating multiple dietary compositions, this study builds upon existing knowledge by exploring the potential benefits of diverse bran sources and determining optimal feed formulations.

Comment 3 The hypothesis or specific research questions are not explicitly stated. A clear hypothesis would improve focus.

Response Thank you for your feedback. I have now explicitly stated the hypothesis in the text.

Methodology

Comment 4 The study lacks a clear justification for the sample size. How was the number of mealworms per treatment determined? Was there a power analysis?

Response Thank you for raising this important point. The sample size of mealworms per treatment was determined based on preliminary studies and existing literature that suggest sufficient replication is needed to achieve reliable results. Each replicate contained fifty larvae, randomly transferred to a plastic container (22 × 28 × 15 cm) with aeration slits on the sides.

Refrences: Van Broekhoven, S., Oonincx, D. G., Van Huis, A., & Van Loon, J. J. (2015). Growth performance and feed conversion efficiency of three edible mealworm species (Coleoptera: Tenebrionidae) on diets composed of organic by-products. Journal of insect physiology, 73, 1-10.‏

Comment 5 The experiment lasted 12 weeks, but there is no discussion on whether this timeframe is sufficient to assess long-term dietary effects.

Response Thank you for your comment. The 12-week experimental duration was chosen based on the larval life cycle of T. molitor, which typically spans about three months, and aligns with established literature demonstrating that this timeframe is sufficient to evaluate growth and developmental outcomes. Additionally, prior studies have shown that initiating growth assessments four weeks into the larval stage provides reliable insights into dietary impacts. Thus, the 12-week period is appropriate for capturing significant dietary effects on growth performance and nutritional composition.

Refrences:

van Broekhoven S, Oonincx DGAB, van Huis A, van Loon JJA. Growth performance and feed conversion efficiency of three edible mealworm species (Coleoptera: Tenebrionidae) on diets composed of organic by-products. Journal of Insect Physiology. 2015;73:1-10.‏

Bjørge JD, Overgaard J, Malte H, Gianotten N, Heckmann L-H. Role of temperature on growth and metabolic rate in the tenebrionid beetles Alphitobius diaperinus and Tenebrio molitor. Journal of Insect Physiology. 2018;107:89-96.

Comment 6 Potential biases: Since mealworm growth can be influenced by factors such as rearing density, genetic variability, and microclimate conditions, were these controlled?

Response Thank you for your comment. To minimize potential biases, we controlled for key influencing factors: rearing density was standardized across all treatments, mealworms were sourced from a genetically homogeneous population, and microclimate conditions (temperature, humidity, and light) were maintained consistently throughout the experiment.

Comment 7 The moisture addition (carrots) could introduce variation. Was this accounted for in data interpretation?

Response Thank you for your comment. The moisture content from carrots was carefully monitored and standardized across all treatments to minimize variability. For moisture, 2 grams of fresh carrot was added twice a week, and old carrot pieces were removed. Any uneaten carrot was dried at 100°C until reaching a constant weight. The dried weight was compared to the dry weight of a carrot piece with the same initial fresh weight, ensuring accurate moisture content adjustments in the analysis.

Results

Comment 8 Some results la

---

## [Decision Letter · Decision Letter 1]

Influence of dietary composition on the nutritional profile and feed conversion efficiency of Tenebrio molitor

PONE-D-25-04529R1

Dear Dr. Akmali,

We’re pleased to inform you that your manuscript has been judged scientifically suitable for publication and will be formally accepted for publication once it meets all outstanding technical requirements.

Kind regards,

António Raposo

Academic Editor

PLOS ONE

Additional Editor Comments (optional):

Reviewers' comments:

Reviewer's Responses to Questions

**Comments to the Author**

Reviewer #4: All comments have been addressed

Reviewer #5: All comments have been addressed

2. Is the manuscript technically sound, and do the data support the conclusions?

Reviewer #4: Yes

Reviewer #5: Yes

3. Has the statistical analysis been performed appropriately and rigorously?

Reviewer #4: No

Reviewer #5: Yes

4. Have the authors made all data underlying the findings in their manuscript fully available?

Reviewer #4: No

Reviewer #5: Yes

5. Is the manuscript presented in an intelligible fashion and written in standard English?

Reviewer #4: No

Reviewer #5: Yes

Reviewer #4: Dear Author, your ms it is not at the level of the magazine, I have objections on a scientific level, n=3 is not a number of insects that corresponds to research at this level. I apologize, unfortunately I will have to reject the your ms.

Reviewer #5: (No Response)

**Do you want your identity to be public for this peer review?** For information about this choice, including consent withdrawal, please see our Privacy Policy

Reviewer #4: No

Reviewer #5: No

---

## [Editor Report · Acceptance letter]

PONE-D-25-04529R1

PLOS ONE

Dear Dr. Akmali,

I'm pleased to inform you that your manuscript has been deemed suitable for publication in PLOS ONE. Congratulations! Your manuscript is now being handed over to our production team.

Kind regards,

on behalf of

Dr. António Raposo

Academic Editor

PLOS ONE